# Surface Modification of Titanate Nanotubes with a Carboxylic Arm for Further Functionalization Intended to Pharmaceutical Applications

**DOI:** 10.3390/pharmaceutics15122780

**Published:** 2023-12-15

**Authors:** Ranim Saker, Orsolya Jójárt-Laczkovich, Géza Regdon, Tamás Takács, Imre Szenti, Noémi Bózsity-Faragó, István Zupkó, Tamás Sovány

**Affiliations:** 1Institute of Pharmaceutical Technology and Regulatory Affairs, University of Szeged, Eötvös u 6., H-6720 Szeged, Hungary; rnmsaker@gmail.com (R.S.); jojartne.laczkovich.orsolya@szte.hu (O.J.-L.);; 2Department of Applied and Environmental Chemistry, University of Szeged, Rerrich Béla tér. 1., H-6720 Szeged, Hungary; takacstamas@chem.u-szeged.hu (T.T.); szentiimre@gmail.com (I.S.); 3Institute of Pharmacodynamics and Biopharmacy, University of Szeged, Eötvös u 6., H-6720 Szeged, Hungary; bozsity-farago.noemi@szte.hu (N.B.-F.); zupko.istvan@szte.hu (I.Z.)

**Keywords:** titanate nanotubes, carboxylic acid, polyethylene glycol, functionalization, linkers

## Abstract

Nanotechnology is playing a significant role in modern life with tremendous potential and promising results in almost every domain, especially the pharmaceutical one. The impressive performance of nanomaterials is shaping the future of science and revolutionizing the traditional concepts of industry and research. Titanate nanotubes (TNTs) are one of these novel entities that became an appropriate choice to apply in several platforms due to their remarkable properties such as preparation simplicity, high stability, good biocompatibility, affordability and low toxicity. Surface modification of these nanotubes is also promoting their superior characters and contributing more to the enhancement of their performance. In this research work, an attempt was made to functionalize the surface of titanate nanotubes with carboxylic groups to increase their surface reactivity and widen the possibility of bonding different molecules that could not be bonded directly. Three carboxylic acids were investigated (trichloroacetic acid, citric acid and acrylic acid), and the prepared composites were examined using FT-IR and Raman spectroscopy, scanning electron microscopy (SEM), transmission electron microscopy (TEM) and dynamic light scattering (DLS). The toxicity of these functionalized TNTs was also investigated using adherent cancer cell lines and fibroblasts to determine their safety profile and to draw the basic lines for their intended future application. Based on the experimental results, acrylic acid could be the suitable choice for permanent surface modification with multiple carboxylic groups due to its possibility to be polymerized, thus presenting the opportunity to link additional molecules of interest such as polyethylene glycol (PEG) and/or other molecules at the same time.

## 1. Introduction

Nowadays, an accelerated transformation procedure from the use of traditional bulk materials into applying smart, predesigned nano ones is taking place in all aspects of modern life. All scientific disciplines should be keeping up with this tremendous evolution, and every single domain should explore the potential of investing in others’ invented tools for its own benefit.

Titanate nanotubes (TNTs) are one of these newly invented materials that were introduced as efficient tools to serve in various fields aiming to achieve different purposes. Several research works demonstrated the effectiveness of TNTs in medical, industrial and chemical disciplines such as in dental implants and orthopedics [1,2], chemical catalysis, biofuel synthesis, solar cells and water purification [3,4,5,6]. This outstanding performance in these multiple fields brings out an insistent question about the potential role they could play in the pharmaceutical domain.

For this reason, TNTs are attracting more and more attention in the pharmaceutical community as they could offer a novel platform and an appealing candidate for pharmaceutical uses, especially in the field of drug delivery. Their attractiveness arises from the preferable properties they could offer such as advantageous tubular geometry, large specific surface area compared to their spherical counterparts, hydrophilic nature, nano-size undetectable for the reticuloendothelial system, good biocompatibility and mechanical strength. This unique package of properties is of fundamental importance and is considered a key feature for their prospective use as drug carriers [1,7,8].

Unfortunately, only a few attempts have been made to bring these nanomaterials into therapeutic practice. For instance, promising results were successfully obtained in the field of oncology after using TNTs as antitumor carriers for doxorubicin and docetaxel to pancreatic and prostate tumors [9,10]. In fact, just depending on their photocatalytic activity, TNTs could induce the death of malignant cells upon UV irradiation [11]. This ability holds the potential of introducing a synergetic effect if combined with the loading of active pharmaceutical ingredients (APIs). In this context, TNTs were investigated as possible nanocarriers for various APIs such as ibuprofen, dexamethasone, curcumin, methotrexate and silibinin [12,13,14,15]. The use of TNTs as drug carriers would not just contribute to the effective therapeutic outcomes after the drug has been delivered to the site of action, but it would also have advantages on the technological and industrial levels. According to Sipos et al., the incorporation of APIs into TNTs could present superior processability in comparison to APIs alone, and this would shorten the time needed for preformulation studies, especially regarding the tableting process [16,17,18].

Although, the mentioned properties of TNTs are considered as advantages for several applications including medical and pharmaceutical ones. Nevertheless, surface modification could be an unavoidable step for further improvement of the existing properties, for adjustment of the unfavorable ones or even for providing them with new characteristics. The surface modification is considered as a recommended solution to overcome TNTs problems and upgrade their use to new scopes; for example, applying a hydrophobic cap for preventing the leaching of hydrophilic drugs to the surrounding environment and achieving a controllable release profile [19] or for enhancing the permeability of these hydrophilic nanotubes and facilitating their absorption/biological membrane bypassing [20]. Surface modification of TNTs was also conducted successfully as an approach to enhance the APIs loading, sustain/control the release rate and reduce toxicity using different molecules such as polyamidoamine dendrimers, P25 nanoparticles and chitosan [14,15,21,22].

However, it is worth mentioning that while there is a considerable number of scientific papers discussing surface functionalization of different nanomaterials, just a few of them focused on this type of modification with TNTs. This lack of research in this fundamental angle opens the door for further investigation in this unexplored subject.

This study aims to contribute to TNTs surface functionalization by creating an extended and flexible arm on their surface using carboxylic acids. This proposed approach could be an appropriate way to create functional nanomaterials, facilitating further functionalization with additional molecules, and would result in tailoring TNTs surface chemistry to enhance their characters or adapt them to fit well with the pharmaceutical requirements. Polyethylene glycol (PEG) was selected as the functionalizing agent that will be bonded to the carboxylic arm due to its well-known biological benefits such as resisting non-specific protein adsorption and thus the formation of protein corona [23,24,25], prolonging circulation time [26] and reducing cytotoxicity [27].

Creating a functional arm on the surface of TNTs for special purposes has been carried out before by Marques et al. to enhance TNTs adsorption capacity towards anionic materials in aqueous solutions. This purification technique could be achieved by linking amine groups to the originally negative-charged TNTs surface. These amine groups could turn into positively charged groups after protonation in acidic media, leading to electrostatic interactions with negatively charged materials. Furthermore, the unfunctionalized sites of TNTs surface could continue to interact with cationic materials as these sites preserve their original negative charge. According to this strategy, TNTs could act as a bi-charged surface and interact with molecules that could not interact with TNTs in their original form [28].

A similar approach was suggested in this research article (Figure 1) which involves enriching TNTs surface with reactive carboxylic groups that could facilitate subsequent grafting of numerous molecules intended for shaping the surface properties or, furthermore, attaching biologically active substances that could not be directly linked to the original surface of TNTs. This proposed idea to create a functional carboxylic arm on the surface of TNTs is easily applicable with the organic counterpart through oxidation [29], which will provide carbon nanotubes (CNTs) with carboxylic groups, giving them the opportunity to be covalently functionalized [30]. However, with inorganic nanomaterials, this task is more challenging and more difficult to achieve. This study will also thoroughly discuss the type of the created interactions, as the desired one would differ according to the intended future implementation. To the best of our knowledge, this is the first paper to discuss the importance of tailoring the surface chemistry of nanotubes and create a connecting bridge with carboxylic groups that could serve as future linkers for additional molecules (APIs, markers, etc.) and then to investigate the toxicity of these modified samples in order to upgrade their use for safe therapeutic implementation.

## 2. Materials and Methods

### 2.1. Materials

Three types of carboxylic acids, trichloroacetic acid (TCA), citric acid (CA) monohydrate (both from Molar, Chemicals Ltd., Budapest, Hungary) and acrylic acid (AA) (Sigma-Aldrich Ltd., Budapest, Hungary), in addition to two types of polyethylene glycol (PEG 600 and PEG 6000) (Fluka AG, Buchs, Switzerland), were used to functionalize the surface of TNTs. Sodium persulfate, sodium sulfite and sodium hypophosphite monohydrate reagents were also purchased from Sigma-Aldrich Ltd. (Budapest, Hungary).

### 2.2. Methods

#### 2.2.1. Preparation of TNTs

TNTs were prepared at the Department of Applied and Environmental Chemistry, University of Szeged, according to Sipos et al. [18], using the hydrothermal treatment method. Briefly, 120 g of NaOH was added to 300 mL of distilled water during continuous mixing, then 75 g of TiO_2_ was added and mixed for 15 min. This mixture was put in an autoclave at 185 °C for 24 h then cooled at room temperature for 2 h, followed by cooling with cold water. The TNTs were finally washed with distilled water under vacuum using filter No:4.

#### 2.2.2. Preparation of Carboxylic Acid Functionalized TNTs

TCA functionalization of TNTs was performed by adding 3 g of TNTs to 90 mL of water in an ultrasonic bath for 1 h until a homogenous suspension was obtained. This suspension was heated at 80 °C in a condenser connected to nitrogen gas for 30 min, then TCA was added to the previous system and mixed for one day.

CA functionalized TNTs (CA-TNTs) were prepared by adding 0.5 g of TNTs to 15 mL of water containing citric acid on a magnetic stirrer. This mixture was stirred for 30 min to obtain a homogenous suspension then heated at 50 °C with continuous stirring for 24 h.

AA functionalized TNTs (AA-TNTs) were prepared by mixing 1 g of TNTs with 28.8 g of acrylic acid, 32 g of hexane and 8 g of water then placed for sonication in an ultrasonic bath for 20 min at room temperature. This mixture was stirred at room temperature in a magnetic stirrer for 2 days then separated by centrifugation at 12,000 rpm for 60 min at 8 °C.

#### 2.2.3. Functionalization with PEG

CA-TNTs were further functionalized with PEG 600 by mixing CA-TNTs and PEG 600 at 130 °C in a silicon oil bath for 24 h in an inert atmosphere condition by bubbling nitrogen gas, then the mixture was cooled to room temperature and filtered.

AA/PEG 6000 functionalized TNTs were prepared by applying a two-step method, where first an AA-PEG copolymer was prepared according to Abo-shosha et al. [31] and Ibrahim et al. [32], followed by bonding the copolymer with TNTs. Briefly, the AA-PEG 6000 copolymer was prepared by a polymerization reaction in a thermostatic water bath at 40 °C under atmospheric oxygen. The polymerization medium of polyacrylic acid (PAA)/PEG was prepared by neutralizing 20% of AA (115.5 g) with the equivalent amount of an aqueous solution of 500 g/L of NaOH followed by dissolving 40 g of PEG 6000. After that, 3.3 mL of 40 g/L sodium sulfite (Na_2_SO_3_) solution and 12.2 mL of 330 g/L sodium persulfate (Na_2_S_2_O_8_) solution were added with stirring. An exothermic polymerization reaction commenced after an induction period with the evolution of water vapor, followed by solidification of the polymerization medium. The latter was then cooled, disintegrated, oven dried at 105 °C for 2 h, cooled and kept over silica gel for at least 40 h before analysis. In the second step, this copolymer (50 g/L) was bonded with TNTs (5 g/L) after adding Na-hypophosphite (15 g/L) as a catalyst in this reaction.

All the prepared composites were subjected to a washing step to remove any residues adsorbed on the surface of TNTs and finally were dried in a drying oven (Memmert, Büchenbach, Germany).

#### 2.2.4. Structural Investigations

The determination of the success of functionalization procedure and the nature of TNT–carboxylic acid interactions were evaluated using an Avatar 330 FT-IR spectrometer (Thermo Fisher Scientific Ltd., Waltham, MA, USA). FT-IR measurements were conducted with a transmission E.S.P. accessory by using 128 scans at a resolution of 4 cm^−1^ and applying H_2_O and CO_2_ corrections. Spectragryph 1.2.16.1 software (Friedrich Menges, Obersdorf, Germany) was used to evaluate the results.

A DXR Dispersive Raman spectrometer (Thermo Fisher Scientific Inc., Waltham, MA, USA) equipped with a CCD camera and a diode laser operating at a wavelength of 780 nm was applied to perform Raman measurements, which were carried out with a laser power of 12 and 24 mW at 25 μm slit aperture size. The data were collected in the spectral range of 200–3300 cm^−1^ using photobleaching to compensate fluorescence of titanate. OMNIC 8 software was used for data collection, averaging the total of 20 scans and making the spectral corrections. For the removal of cosmic rays, a convolution filter was applied on the original spectrum using Gaussian kernel.

#### 2.2.5. Morphology

The morphology and size of bare and functionalized TNTs were investigated by scanning electron microscope (SEM) (Apreo C, Thermo Fisher Scientific Ltd., Waltham, MA, USA) and transmission electron microscope (TEM) (FEI Tecnai G2 20 X-TWIN, Hillsboro, OR, USA). SEM was instrumented with a cold field emission (CFE) cathode. The system was used under 10^−7^ Pa ultra-high vacuum, and the samples were maintained at room temperature and under 10^−4^ Pa vacuum during the characterization in the sample chamber. TEM images were taken at 200 kV of electron energy.

The hydrodynamic diameter and zeta potential measurements were taken using dynamic light scattering with a Nano ZS zetasizer system (Malvern Panalytical Ltd., Malvern, Worcestershire, UK), equipped with a 633 nm wavelength laser. The bare and functionalized TNT samples were prepared by dispersing them in different media (water, phosphate-buffered saline (PBS) and a PBS-based cell culture medium) with 30 min of ultrasonication, and then 1 mL of each dispersion was placed in folded capillary zeta cells.

#### 2.2.6. Cell Viability Studies

The direct toxicity of the newly prepared TNTs was determined on two intact and three cancerous cell lines by standard MTT (3-(4,5-dimethylthiazol-2-yl)-2,5-diphenyltetrazolium bromide) method [33]. The TNTs were tested against non-cancerous human embryo fibroblast (MRC5) and mouse embryonic fibroblast cell lines (NIH/3T3), as well as on malignant human ovarian carcinoma (A2780) and two types of oropharyngeal squamous carcinoma cell lines (UPCI-SCC-131 and UPCI-SCC-154). The MRC5, NIH/3T3 and A 2780 cell lines were purchased from the European Collection of Cell Cultures (ECACC, Salisbury, UK), while the two oropharyngeal cell lines from the German Collection of Microorganisms and Cell Cultures GmbH (Braunschweig, Germany). Each cell line was maintained at 37 °C in a humidified atmosphere (containing 5% CO_2_) in Eagle’s Minimum Essential Medium (EMEM) supplemented with the appropriate amount of heat-inactivated fetal calf serum (FCS), non-essential amino acids and 1% antibiotic–antimycotic mixture (penicillin–streptomycin), according to the manufacturer’s recommendations. All media and supplements were obtained from Lonza Group Ltd. (Basel, Switzerland).

For testing the action on cell growth, cells were seeded into 96-well plates at the density of 5000 cells/well, and after overnight standing, cells were treated with increasing concentrations of TNTs (1, 3, 10 and 30 µg/mL). After incubation for 72 h, 5 mg/mL MTT solution was added for another 4 h. The precipitated formazan crystals were solubilized in dimethyl sulfoxide, and the absorbance was measured at 545 nm using a microplate reader (SPECTROstarNano, BMG Labtech GmbH, Offenburg, Germany). Wells with untreated cells were utilized as control. The in vitro experiments were carried out twice with five parallels. TNTs were suspended in dimethylsulfoxide (DMSO), and the highest DMSO content of the medium (0.6%) did not substantially affect cell proliferation. Data were evaluated with GraphPad Prism version 10 for Windows software (GraphPad Software, San Diego, CA, USA), while statistical evaluation was performed with TIBCO Statistica v14.0.1.25 software (TIBCO Software Inc., Palo Alto, CA, USA).

## 3. Results and Discussion

Previous attempts to directly link PEG to TNTs were not successful. Although PEG was successfully linked to TNTs via H-bonds, the strength was not sufficient to hold the complex together after dispersion in aqueous medium, so the PEG was rapidly detached from the TNTs. Therefore, the main idea of creating a carboxylic arm on the surface of TNTs is to serve as a functional bridge and enable strong, covalent connecting of various molecules that could not be directly connected to TNTs hydroxylated surface.

In the first attempt, the same method which was previously used for functionalization of TNTs with trichloro-octyl-silane [20] was adopted to TCA. Nevertheless, the trial was not successful, as shown in Figure 2, as the characteristic peaks of TCA, mainly the C=O, C-O and C-Cl stretching vibration at 1751, 1262 and 679 cm^−1^, respectively, are clearly absent in the spectrum of the corresponding composite, which suggests that it was removed during the washing procedure. This unsuccessful attempt could be attributed to the fact that TCA has only one carboxyl group, which was apparently not sufficient for creating a durable association with the surface of TNTs.

In contrast to the previous results, the functionalization experiments with citric acid appeared to be successful, and this was evidenced by the existence of its characteristic peaks (C=O stretching at 1728 cm^−1^ and C-O stretching with two peaks at 1233 cm^−1^ and 1213 cm^−1^, belonging to CHOH and CH_2_ attached carboxyl groups, respectively) in the spectra of CA-TNTs (Figure 3). However, the slight shift of the C=O stretching to 1717 cm^−1^ and the CH_2_-C-O to 1200 cm^−1^ indicates a change in association due to the establishing of hydrogen bonds between the two entities, while the peak at 1233 cm^−1^ remained in an unchanged position. This suggests that the two primary COOH are attached to the surface of TNTs, while the third one is facing outwards in an unchanged form.

This would enable further functionalization trials with additional molecules intended to be connected to the free carboxylic arm. An esterification attempt between the carboxylated surface of CA-TNTs with PEG was made depending on the theory that titanate could serve as a heterogenous catalyst for such chemical esterification reactions [34,35]. For this reason, TNTs were used as a part of the reaction in addition to using them as a potential catalyst.

Unfortunately, in this study this theory did not work, and the planned esterification reaction failed, as was confirmed by the results of FT-IR examinations (Figure 4). The reaction outcome was purified with filtration. The spectrum of the filtered precipitate contained only the signals of the CA-TNTs. In addition, the unique peaks that are attributed to PEG structure, especially the C-H stretching vibration at 2868 cm^−1^ and the C-O stretching vibration at 1102 cm^−1^, are absent, while the dried filtrate provided an identical spectrum of PEG. It could be concluded from the previous observations that PEG was not bonded successfully to the carboxylated surface of TNTs as it was probably removed with water during the washing step.

According to these results, the choosing of multifunctional group carboxylic acid as citric acid is more favorable in terms of using one group to interact with TNTs surface and leaving the other one or two for further interactions with additional molecules. Nevertheless, it also could be concluded that creating a weak interaction/association between the surface of TNTs and the carboxylic acids is not sufficient to step further for additional functionalization, as the subsequent treatment could easily break this fragile association and result in bare TNTs after carboxylic acid is detached off the surface. This type of weak interaction, such as electrostatic attractions and hydrogen bonds, could be more favorable regarding TNTs loading with drugs so these drugs can be liberated later from this association and released to the biological medium. It is worth mentioning that these interactions could slow the release rate of the drug or change its kinetic if they were sufficiently intense [18,36]. On the other hand, stronger bonds are more favorable during functionalization/surface modification, so this surface adjustment could be permanent to provide the TNTs with special specifications intended for special purposes.

As the attempt for TNTs PEGylation via CA linker has failed, a third attempt using AA was applied where the hypothesis was to bond AA to TNTs by π-bonding via breaking up of its C=C bond. Although a weak interaction with the surface of TNTs was achieved similarly as the case with CA, the attempt was considered a failed experiment. Since the characteristic peaks of AA (C=O stretching vibration at 1722 cm^−1^, the OH bending at 1410 cm^−1^, and the C-O stretching at 1299 cm^−1^) exhibited significant shifting (1630, 1440 and 1277 cm^−1^, respectively) in the spectrum of the reaction result, the C=C stretching vibration at 1636 cm^−1^ was covered while =C-H bending vibration at 986 cm^−1^ remained in an unchanged position (Figure 5). These observations indicate that instead of the π-bonding with the C=C bond, the carboxylic group formed an H-bond with the TNTs. In this case, the carboxyl group of acrylic acid would be occupied and thus would not be available to act as an extended arm for further functionalization. Moreover, as it was previously mentioned, such a weak interaction as H-bonds with acrylic acid is not enough to step further and use this acid as a linker for additional molecules.

Based on the previous discussion, it is necessary to build a durable connection between the surface of TNTs and the carboxylic acid so it could tolerate the subsequent treatment for using it as a bridge connecting additional substances. For this reason, a reverse approach was conducted starting from bonding the acrylic acid with the molecule of interest (PEG) as the initial step depending on the carbon double bond as the main site of interaction. Then, this synthesized adduct could be linked to the hydroxylated surface of TNTs. Acrylic acid–PEG connection was carried out using free radical polymerization reaction, which will present a new copolymer (PAA–PEG) with multiple carboxylic groups. These groups could be utilized later to connect more molecules with the carboxylic arm of the pegylated TNTs such as drugs, markers or biological molecules.

Significant differences may be observed between the spectrum of the prepared copolymer and its precursors (PEG and AA) spectra (Figure 6). The disappearance of the peak assigned to the vibration of H in C=C-H at 986 cm^−1^ is clear evidence of the polymerization reaction. Furthermore, the absorption peak of C-H stretching of methylene groups is also visible at 2928 cm^−1^ in the spectrum of the prepared copolymer, and it also could be a sign for the saturation of C=C bond in acrylic acid during the polymerization process. Regarding the C=O stretching range between 1600 and 1750 cm^−1^, there were two peaks visible in the spectrum of acrylic acid assigned to C=C and C=O vibrations which are at 1636 and 1727 cm^−1^, respectively. In the newly synthesized material, the peak at 1727 cm^−1^ is separated to 1715 and 1737 cm^−1^, indicating the presence of terminal and mid-chain carboxyl groups. The new peaks at 1558 and 1538 cm^−1^ may indicate that the carboxyl groups are presented in the form of carboxylate anions. The slight decrement of the O-H bending vibration of PEG at 1349 cm^−1^ and the increasement of C-O stretching at 1245 cm^−1^ in the copolymer indicate that PEG is attached to the end of the PAA chain through its terminal OH.

Regarding the spectrum of AA/PEG functionalized TNTs, the further shift of C=O stretching to 1706 cm^−1^, and the multiple shifts of the peaks in the C-O stretching region at 1171, 1131 and 1038 cm^−1^, indicate that the copolymer is strongly attached to the surface of TNTs. A similar slight but considerable shift may be observed in the carboxylate stretching to 1560 and 1540 cm^−1^. According to Pouran et al. [37] and Liew et al. [38], this shift indicates a strong ionic bond between the carboxylate anion and Ti ions which might be irreversible.

The results were cross-confirmed by Raman spectroscopic investigations. It is well visible on Figure 7 that the intensive C=C stretching signal of AA at 1638 cm^−1^ has completely disappeared from the AA/PEG copolymer spectrum, confirming the successful free radical polymerization reaction. The other characteristic peaks at 1452 cm^−1^, 1726 cm^−1^ and 2931 cm^−1^ may be attributed to the stretching of carboxylate anion, carbonyl groups and CH_2_ groups, respectively. The peak at 2931 cm^−1^ may be clearly identified in the spectrum of AA/PEG functionalized TNTs in an unchanged position, but the other signals cannot be identified clearly due to the fluorescence of TNTs. However, a new peak with small intensity has appeared in the spectrum at 1970 cm^−1^, which may also indicate the presence of intermolecular interactions that may be due to the conjugated Ti-O-C vibrations.

Morphological investigations have also been conducted on the bare and functionalized samples, with both scanning and transmission electron microscopy (Figure 8). No considerable difference was observed between the bare and functionalized TNTs except for some fragmentation, which may be due to the ultrasonication which was to aid the dispersion of TNTs before the functionalization reaction. The diameter was found to be 9.56 ± 1.56 nm vs. 10.42 ± 2.03 nm, while the length was found to be 128.27 ± 61.22 nm vs. 92.54 ± 20.97 nm for bare vs. functionalized TNTs, respectively, according to TEM pictures.

Nevertheless, these results show no correlation with those obtained with DLS measurements, where the hydrodynamic diameter in water was found to be 212.7 ± 0.2 nm vs. 417.2 ± 10.1 nm for bare and functionalized TNTs, respectively. The larger average particle size in water in comparison with that in the dried state can be attributed to the existence of a hydration layer that surrounds the hydrophilic nanotubes due to their hydroxylated surface, while the increased size of functionalized TNTs may be further proof of the successful functionalization, as the presence of AA/PEG copolymer on the surface of TNTs may lead to greater size and increased interaction with water molecules, probably due to their entrapment between its chains.

The hydroxylated surface of TNTs provides a stable negative zeta potential (−36.47 ± 0.35 mV), which corresponds well with the results of Bavykin et al. (−42.7 mV) [39] and of Papa et al., who reported a strongly negative zeta potential above the isoelectric point (pH 3.6) of TNTs [40], and indicated that although these nanotubes would be negatively charged at the physiological pH (−34.5 ± 4.3 mV), they can still interact well with the negatively charged cell surface, achieving good internalization supported by their tubular morphology [41]. The functionalized TNTs exhibited the same stable negative zeta potential in purified water (−36.86 ± 0.41 mV).

However, there was a considerable difference in the behavior of bare and functionalized TNTs in PBS buffer where, despite the unchanged zeta potential (−33.96 ± 2.27 mV and −35.9 ± 2.01, respectively), the bare TNTs exhibited visible precipitation, which was in accordance with the size measurements that showed a size of 11,037 ± 7399 nm. According to our hypothesis, the PBS buffer induced a partial ion exchange on the surface of TNTs, which decreased the repulsion forces and induced the aggregation of nanotubes. In contrast, no similar effect was observed in the case of the functionalized TNTs, where only a slight increment was observed in the hydrodynamic diameter (570.43 ± 26.53 nm), so functionalization here has a positive impact by reducing agglomeration and enhancing dispersibility and stability of TNTs in this medium.

Similar observations were detected in the PBS-based cell culture medium, where 11,283 ± 6480 nm and 837.83 ± 18.48 nm size was detected for bare and functionalized TNTs, respectively. The increased size in the cell culture medium can be due to the decreased zeta potential (−3.14 ± 1.79 mV and −23.2 ± 0.95, for bare and functionalized TNTs, respectively) indicating strong TNT–cell interactions. However, another possible explanation is that the presence of cells probably interferes with the measurement and might influence the results. Nevertheless, the smaller change in zeta potential of the functionalized samples in cell culture medium compared to that of the bare TNTs also supports the stabilizing effect of functionalization, probably by sterically hindering the hydrophobic and electrostatic interactions with the cells, reducing their adsorption on the surface, and probably simulating what could happen with the plasma components [42]. However, decreasing the negative zeta potential would result in reducing the repulsion with cell membrane, thus leading to the enhancement of cell permeation, higher penetration but also higher potential of toxic effects [43].

The preliminary toxicity of the prepared TNTs has been investigated by a widely used viability assay against human adherent cancer cell lines and two different fibroblasts. The substances were applied in 1–30 μg/mL concentration range for 72 h. None of them exerted considerable cell growth inhibition as the growth inhibition values did not exceed 30% in any cases, so the investigated samples can be considered as safe (Figure 9). The statistical analysis revealed no significant (*p* > 0.05) difference between the tested samples. Regarding the sensitivity of various cell lines, the statistical evaluation confirmed that UPCI-SCC-154 oral squamous carcinoma cell line is the most sensitive cell line but in a non-concentration-dependent manner. Typical concentration-dependent effects were observed only against UPCI-SCC-131 cells where the 30 μg/mL concentration exhibited a significant increase in the growth inhibition in comparison with the 1 and 3 μg/mL concentrations (*p* = 0.012 and *p* = 0.011, respectively), but the inhibition did not exceed the 30% level even in this case. The proliferation of ovarian cancer cell line A2780 and fibroblasts were practically not influenced by the treatment with TNTs. There were no significant (*p* > 0.05) differences between human and murine fibroblasts (MRC5 and NIH/3T3, respectively) concerning the sensitivity toward the nanotubes. Though the presented cell-based viability results cannot substitute an appropriate toxicological study, they indicate that the tested substances have no relevant action on the cell growth, and, therefore, no outstanding toxicity can be expected from later in vivo experiments.

These results agreed with the findings of Papa et al., who reported no cytotoxicity of TNTs on Chinese hamster ovary cell lines (CHO) after 24 h of exposure in a concentration up to 10 µg/mL [40]. They were also in agreement with the findings of Fenyvesi et al. [44] and Wadha et al. [45], who demonstrated no cytotoxic effect of TNTs against Caco-2 cell lines in a concentration up to 5 mg/mL after short treatment (120 min) and against A549 epithelial cell lines in a concentration up to 1.1 mg/mL after long exposure (7 days), respectively. It is worth mentioning that these safety results were not limited to the bare TNTs, but the safety was also evidenced with different types of functionalization and with different chemical structures. For example, Ranjous et al. have reported the safety of silan-functionalized TNTs up to 1 mg/mL and stearate-functionalized TNTs up to 2 mg/mL on Caco-2 cell lines using MTT assay for a short exposure (120 min) [20]. Papa et al. have also modified the surface of TNTs with polyethyleneimine (PEI) and then examined the toxicity with MTT assay on cardiomyocytes for 24 and 96 h. No significant toxicity was observed neither with TNTs nor with their functionalized samples in a concentration up to 10 µg/mL [41]. These studies, in addition to our research work, presented promising results to promote the future use of TNTs as safe nanocarriers and a novel platform for drug delivery.

## 4. Conclusions

TNTs have been widely used in different disciplines since they first appeared in 1996 as a superior replacement for the organic carbon counterparts. They hold huge potential that could bring advantages to multiple platforms, including the pharmaceutical one. However, some of their characteristics which act as an obstacle hindering their successful use in health domains could be overcome by functionalizing their surface to shape their properties for suiting the pharmaceutical requirements.

Determining the type and the strength of the created interaction/association between TNTs and the functionalizing agent in the prepared composites is of crucial importance as it would highly affect the subsequent application of these synthesized composites. For instance, weak associations such as hydrogen bonds are not sufficient as the used agent would easily detach off the surface; thus, stronger interactions such as covalent or ionic bonds are more favorable as they are more stable and long-lasting ones.

The synthesized AA/PEG copolymer was successfully utilized for the durable functionalization of TNTs, and it has considerably increased the stability in aqueous environment. Furthermore, based on the results of the preliminary cell viability studies using multiple cell lines, TNTs could be a safe and promising candidate for drug delivery purposes.

## Figures and Tables

**Figure 1 pharmaceutics-15-02780-f001:**
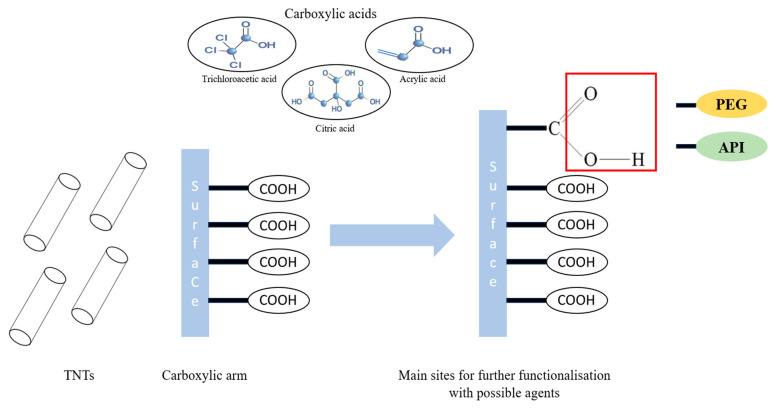
The suggested strategy for TNTs surface modification.

**Figure 2 pharmaceutics-15-02780-f002:**
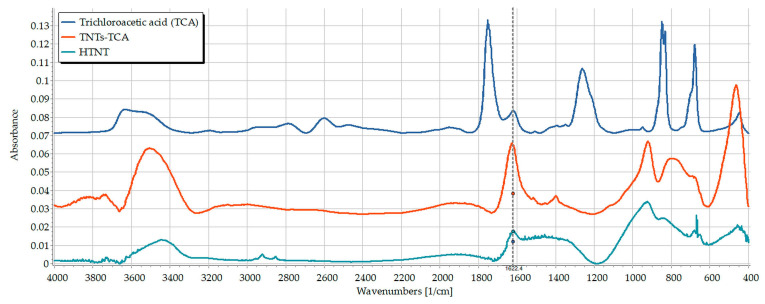
FT-IR spectra of TCA (blue), TNT (cyan) and the TCA-TNTs (red).

**Figure 3 pharmaceutics-15-02780-f003:**
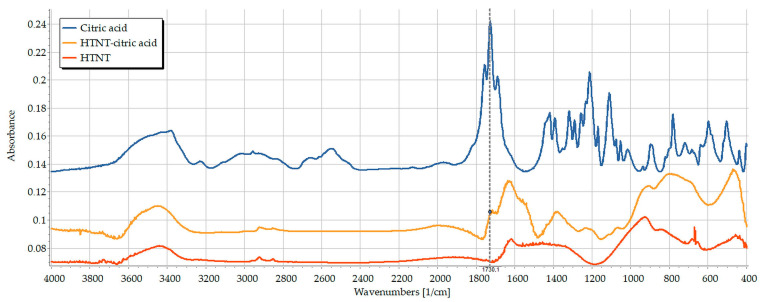
FT-IR spectra of citric acid (blue), TNTs (red) and CA-TNTs (yellow).

**Figure 4 pharmaceutics-15-02780-f004:**
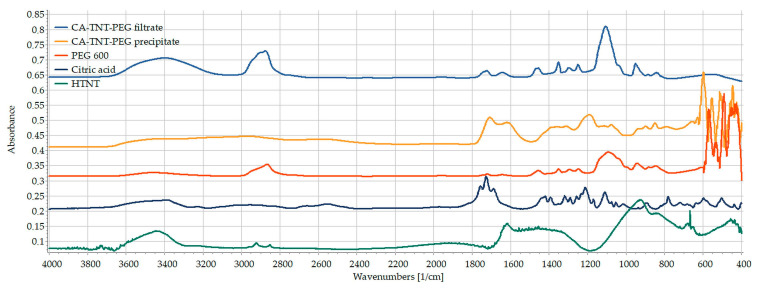
The spectra of PEG 600, esterification result and esterification filtrate.

**Figure 5 pharmaceutics-15-02780-f005:**
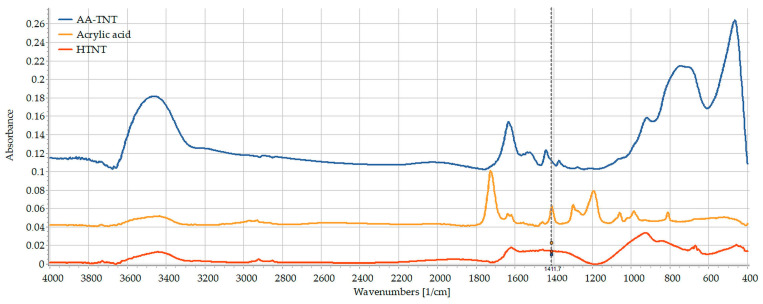
FT-IR spectra of acrylic acid (yellow), TNTs (red) and AA-TNTs (blue).

**Figure 6 pharmaceutics-15-02780-f006:**
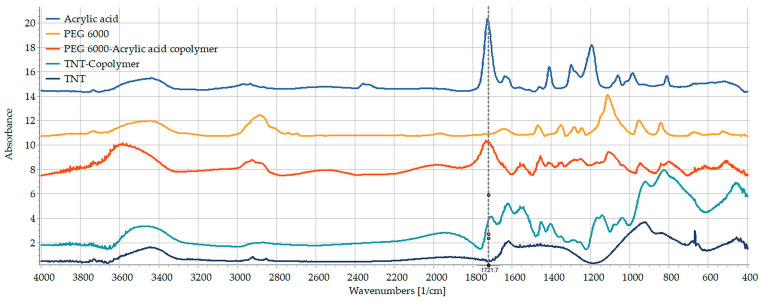
FT-IR spectra of acrylic acid (blue), PEG 6000 (yellow), AA/PEG copolymer (red), TNT (dark blue) and the AA/PEG functionalized TNT (light blue).

**Figure 7 pharmaceutics-15-02780-f007:**
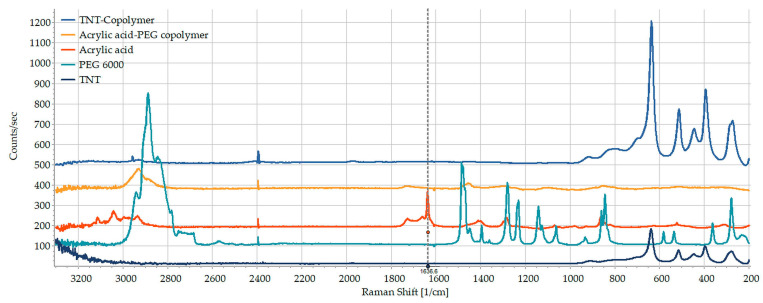
Raman spectra of acrylic acid (red), PEG 6000 (light blue), AA/PEG copolymer (yellow), TNT (dark blue) and the AA/PEG functionalized TNT (blue).

**Figure 8 pharmaceutics-15-02780-f008:**
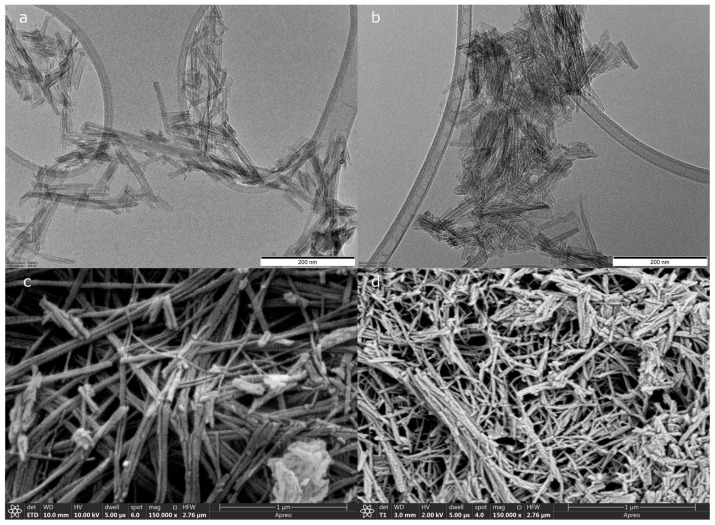
TEM picture of bare (**a**) and AA/PEG functionalized (**b**) TNTs, SEM pictures of bare (**c**) and AA/PEG functionalized (**d**) TNTs.

**Figure 9 pharmaceutics-15-02780-f009:**
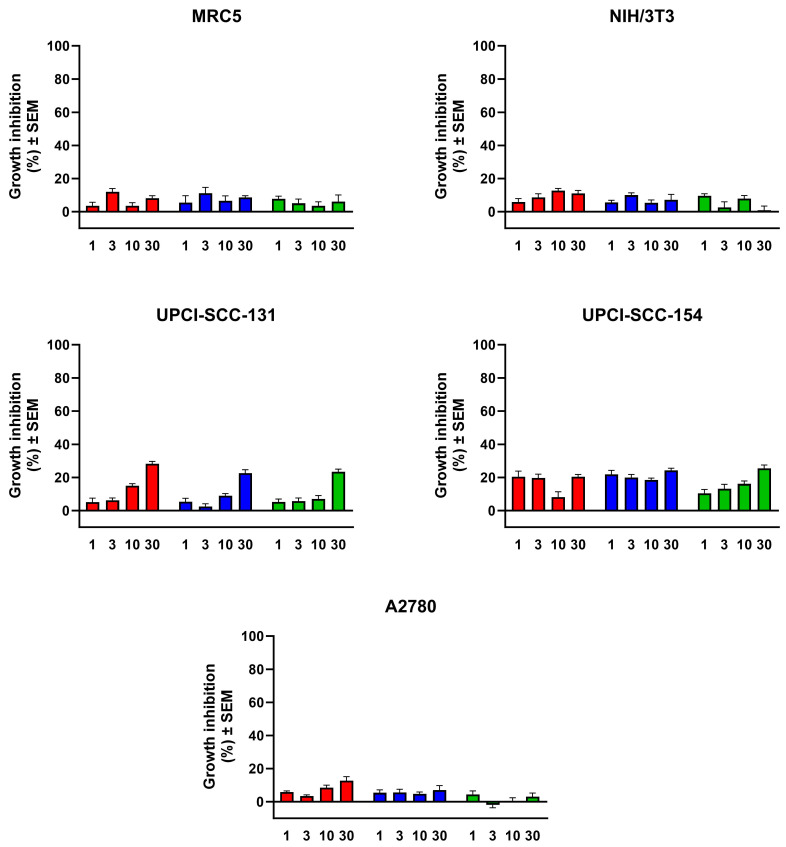
Effects of the prepared nanotubes on the growth of the utilized fibroblasts (MRC5, NIH/3T3) and cancer cells. Numbers on horizontal axes indicate final concentrations in μg/mL; red, blue and green columns mean TNT, AA/PEG copolymer and AA/PEG functionalized TNTs, respectively.

## Data Availability

Data are contained within the article.

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
