# Peer review of "Surface Modification of Titanate Nanotubes with a Carboxylic Arm for Further Functionalization Intended to Pharmaceutical Applications"

_pharmaceutics, 2023, doi:10.3390/pharmaceutics15122780_

Round 1
Reviewer 1 Report
Comments and Suggestions for Authors
The paper entitled „Surface modification of titanate nanotubes with carboxylic arm for further functionalization intended to pharmaceutical applications” describes a modification of the titania nanotubes with three organic acids. Only in the case of acrylic acid, the results were promising, and thus, next studies have been done. Besides IR spectra, the paper contains results from Raman spectroscopy, electron microscopies (TEM and SEM) and preliminary toxicity studies.
The eperimental part is written carefully with satisfactory details. All things related to a ‘chemical kitchen’ are clear. However, some problems should be solved before the publishing of the manuscript:
1. Is it visible in Raman spectra (Fig. 7) that the AA-PEG copolymer is placed on TNT? If yes, it could be slightly more clarified.
2. TNT usually means „titania nanotubes” and I suggest using this term as the explanation of the TNT abbreviation (instead of „titanate nanotubes” used by authors) – the title and lines 41, 95, etc.
3. There is no reference number after „Marques et al.” – line 96
4. Figures 2-7 are just screenshots from the Spectragryph. It is good for a draft, but in the final manuscript it should be prepared more professionally – lines 236, 247, 264, 293, 326, and 340
5. Comment to H-bond with TNTs can be extended with the conclusion that an H-bond is a very weak bond, additionally, so such bonding of AA is also not interesting if we want to use the AA as a linker with TNT – lines 288-291
Comments on the Quality of English LanguageEnglish is good but it still needs a small refinement, e.g. ‘the leaching of hydrophilic drugs to the outsider environment’ – line 76 or ‘two primary COOH is’ – line 245
Reviewer 2 Report
Comments and Suggestions for Authors
Functionalization of oxide nanotubes provides a means to change the surface chemistry of these materials. The authors undertaken an attempt to functionalise the surface of titanate nanotubes with carboxylic groups to increase their surface reactivity and expand the possibility of bonding with different molecules. For this purpose, three carboxylic acids were investigated (trichloroacetic acid, citric acid, and acrylic acid) and the prepared materials were studied by FT-IR and Raman spectroscopy, scanning electron microscopy and transmission electron microscopy. Further the authors determined the toxicity of the new materials toward different types of cells. In general, the paper is interesting but it is lacking the novelty: though the materials were quite different in the chemistry of their surface, as may be judged from FTIR data, they demonstrate no differences in their toxicity. Maybe the authors should try different toxicity tests. Also, XRD data should be included in the manuscript.
Comments on the Quality of English LanguageThe quality of English language is acceptable
Reviewer 3 Report
Comments and Suggestions for Authors
In this study, the authors modified the surface of TNTs using carboxylic acid rich molecules to improve the applicability of TNTs by providing functional groups on the surface to conjugate with various biomolecules. After surface modification, the toxicity of TNTs is examined using in vitro experiments. It is an interesting study, but complete characterization of materials is missing. This study missing detailed examination of the material characterization such as changes in size, surface charge, stability after functionalization. Statistical analysis needs to be performed. Only the shown data is not enough for a publication and will not contribute to the field. So, the reviewer has the following comments to improve the quality of the manuscript.
1. This study lacked detailed examination of the material characterization such as changes in size, surface charge, stability after functionalization. So, the authors are suggested to perform the experiments such as DLS measurements, Zeta potential measurements after and before functionalization.
2. The stability of functionalized TNTS in different aqueous solutions (e.g. PBS, cell culture medium, water), changes in absorption/PL spectra need to be examined.
3. Statistical analysis needs to be performed for the toxicity studies.
Round 2
Reviewer 3 Report
Comments and Suggestions for Authors
The authors satisfactorily addressed the comments of reviewer and improved the manuscript quality. So, it can be accepted for publication.